# Mental Health Nursing Student's Perception of Clinical Simulation about Patients at Risk of Suicide: A Qualitative Study

Pablo Del Pozo-Herce [1,2,3,4], Alberto Tovar-Reinoso [2,3], Antonio Martínez-Sabater [5,6,*],
Elena Chover-Sierra [5,7,*], Teresa Pacheco-Tabuenca [8], Jorge Carrasco-Yubero [3], Juan Luis Sánchez-González [9],
Silvia González-Fernández [10], Iván Santolalla-Arnedo [4,11], Teresa Sufrate-Sorzano [4,11], Raúl Juárez-Vela [4,11]
and Eva García-Carpintero Blas [12]

1 Department of Psychiatry, Fundación Jimenez Diaz University Hospital, 28040 Madrid, Spain; pablo.pozo@quironsalud.es
2 Instituto de Investigación Sanitaria de la Fundación Jiménez Díaz, 28040 Madrid, Spain; alberto.tovar@inv.uam.es
3 School of Nursing Fundación Jiménez Díaz, Madrid Autonomous University, 28040 Madrid, Spain; jorge.carrasco@inv.uam.es
4 Research Unit on Integrated Health Care (INCUiSA), Biomedical Research Center of La Rioja (CIBIR), 26006 Logroño, Spain; ivan.santolalla@unirioja.es (I.S.-A.); teresa.sufrate@unirioja.es (T.S.-S.); raul.juarez@unirioja.es (R.J.-V.)
5 Nursing Care and Education Research Group (GRIECE), GIUV2019-456, Nursing Department, Universitat de Valencia, 46010 Valencia, Spain
6 Care Research Group (INCLIVA), Hospital Clínico Universitario de Valencia, 46010 Valencia, Spain
7 Internal Medicine, Consorcio Hospital General Universitario, 46014 Valencia, Spain
8 Subdirección General de Atención y Cuidados Sociosanitarios, Consejería de Sanidad, 28009 Madrid, Spain
9 Department of Nursing and Physiotherapy, Faculty of Nursing and Physiotherapy, University of Salamanca, 37007 Salamanca, Spain; juanluissanchez@usal.es
10 Faculty of Medicine, University of Salamanca, 37007 Salamanca, Spain; sigofe@usal.es
11 Department of Nursing, Faculty of Health Sciences, University of La Rioja, 26006 Logroño, Spain
12 Health Department, School of Life and Nature Sciences, Nebrija University, 28248 Madrid, Spain; egarcibl@nebrija.es
* Correspondence: antonio.martinez-sabater@uv.es (A.M.-S.); elena.chover@uv.es (E.C.-S.)

**Abstract:** Suicide is a serious public health problem, with a global mortality rate of 1.4% of all deaths worldwide and the leading cause of unnatural death in Spain. Clinical simulation has proven to be a beneficial tool in training nursing students. Such experiences allow them to develop cognitive and affective skills that are fundamental for the detection of warning signs and the use of interventions in cases of people who want to take their own lives. Working in a mental health environment can be difficult for nursing students; therefore, the purpose of this study was to explore the perceptions of nursing students on the approach, management, and intervention of suicidal crisis through clinical mental health simulation. Methods: qualitative descriptive phenomenological study through focus groups and reflective narratives in a sample of 45 students. A thematic analysis was performed using ATLAS-ti. Results: After the analysis, three themes were obtained: (a) management and handling of emotions, (b) identification of suicide motives, and (c) intervention in suicidal crisis. Discussion: Clinical simulation in mental health allows students to exercise clinical judgment reasoning, detect warning signs for a better treatment approach, and provide tools for effective intervention and management of patient care. The results of this study indicate that nursing students face challenges in approaching mental health clinical simulation due to a lack of prior exposure.

**Keywords:** suicide; mental health; simulation training; nursing student; qualitative research

## 1. Introduction

In 1986, the World Health Organization (WHO) defined suicide as "an act with a lethal outcome, deliberately initiated and carried out by the subject, knowing or expecting its lethal outcome and through which he/she intends to obtain the desired changes." [1,2]. Although there is no universal consensus on the terminology of suicidal behavior, related concepts are based on two criteria: "self-inflicted" and "purpose of death" [3,4].

Suicide is one of the leading causes of death in the general population and the third leading cause of adolescent death in the world [5]. It is recognized as a severe public health problem that causes more than 700,000 deaths per year, with figures of eleven suicides per day. These figures are multiplied by 20 when referring to attempts, being the first cause of external death in Spain [6,7], while in the USA, suicide is the tenth leading cause of death in the United States, with more than 47,000 deaths in 2019 [8]. Given its prevalence and global burden, the reduction of suicide deaths has been a global goal of the WHO. It has been included as an indicator (3.4.2. Reduce Suicide Death Rates) within the United Nations Sustainable Development Goals for 2030 [9]. Furthermore, it should be taken into account that, in addition to lethality, suicidal behavior (suicidal ideation, suicide attempt, suicide plan, and death by suicide) produces a negative psychological impact on the personal and close circles of the person who takes their own life, including health professionals [10].

Suicide is a complex phenomenon, influenced by diverse causes at both individual and contextual levels [5,11]. Addressing this problem requires a comprehensive prevention perspective [5] based on multiple interventions at the collective and personal levels that have been changing according to culture, religion, and intellectual principles of history [11,12]. These interventions are aimed at people at risk and imply the need for training of nursing professionals in the management of these behaviors [12,13]. To address suicide risk and prevention, it is crucial to detect risk factors such as the presence of mental disorders [5,11,12,14,15], history of previous suicide attempts [5,12,14,15], advanced age [15], substance abuse [5,16], family problems [14] and conflicting relationships, socio-cultural and economic problems [14], use of psychotropic drugs [15], social isolation [11,12], access to lethal means [15], and hopelessness [12] are the most relevant risk factors since they represent a danger for the development of this behavior, as well as the elements that provide security for life maintenance. In addition, the causes that may precipitate suicidal behavior should be identified.

This comprehensive approach allows for the development of effective strategies such as developing awareness campaigns to reduce the stigma associated with mental health by promoting early detection of signs of risk, implementing follow-up and support programs for those who have previously attempted to take their own lives, ensuring access to mental health services, developing prevention and treatment programs for substance use [13], as well as offering counseling and family support services [12,13,17], fostering social connectedness [11,12] through community programs [12] and stigma reduction [12,18], support groups [10,11,17], implement strategies to reduce access to lethal means such as gun access restrictions and safety education [8] and medication management [13], education of frontline professionals, with particular attention to nurses [10,17,18], addressing the different dimensions of this issue, in addition to brief active contact [17] and digitally driven outreach and interventions [8,13,17]. All of the above contribute to suicide prevention in a more effective way [11,12,17].

Most people who died by suicide had contact with the health care system in the year before their death [8,19], so the nursing professional has a role in suicide prevention by identifying and addressing risk factors and detecting warning signs [20]. However, this can be hindered by lack of training, lack of support, and lack of protocols [21], indicating in studies such as Silva's that 53% of nurses in the United States stated that they had not received prior training in suicide prevention or assessment [10,17,18,22]. Suicide attempts represent a significant clinical challenge in the field of mental health, demanding accurate and effective interventions to prevent future episodes and promote the recovery of affected individuals [13]. The inherent complexity of these cases underscores the need for clinical

training strategies that train nursing professionals to acquire practical assessment and management skills in different situations [19,23].

This training should be included in undergraduate education, and it is essential to provide nursing students with specialized knowledge on early detection of warning signs, suicide risk assessment, communication skills, empathy, active listening, and intervention strategies [10,14]. This training not only enables them to recognize risk and protective factors, but also enables them to provide a supportive, understanding, and caring environment that is essential in the process of recovery and suicide prevention, which can contribute to comprehensive care that is sensitive to the emotional and psychological needs of patients at risk of suicide [16]. Performing a mental status examination and suicide risk assessment is an important skill required of nurses when in the clinical setting. Because nursing students often express anxiety and lack of confidence in doing so, the use of standardized patients provides an excellent opportunity to practice and master this skill in a simulated setting [24].

Overall, simulation has become an essential tool in the field of mental health, as it simulates clinical settings and reveals the potential for advancing this practice. It is an innovative and essential tool for improving clinical competence and decision-making in critical situations [25]. Although simulation offers crucial benefits to clinical training in mental health, its educational development is limited concerning other specialties and faces controversies that hinder its full integration [26]. Extensive research has demonstrated the benefits of incorporating clinical simulation into psychiatric/mental health nursing curricula, emphasizing increased confidence and critical thinking to address the abovementioned difficulties [18]. Significant changes have occurred over the past decade, with many recent experiments and meta-analyses strongly supporting simulation-based mental health education for healthcare professionals [27].

Improving identification and intervention with patients at risk of suicide requires innovative training techniques that safely and effectively teach or enhance the skills of professionals. Advanced clinical simulation, including virtual simulation, allows the transmission of suicide prevention strategies using a realistic and risk-free environment [28,29]. The construction of simulation scenarios provides training in critical situations and the opportunity to discuss and reflect on the topic, increasing awareness and understanding of the issue. It may increase prevention rates [30].

Therefore, the study aims to explore the perceptions of nursing students on the approach, management, and intervention in suicidal crisis through experiences based on clinical mental health simulation.

## 2. Materials and Methods

### 2.1. Study Design

A qualitative descriptive phenomenological study was conducted to explore the perceptions and experiences of students during the clinical simulation of the mental health of patients at risk of suicide, taking into account their circumstances and their points of view [31]. It was carried out according to the theoretical-methodological orientation of Husserl's descriptive phenomenology, which seeks to understand the structure and essence of conscious experience without theoretical prejudices [32]. It focuses on understanding the structure and essence of human experience as it is presented directly in consciousness, without prior theoretical presuppositions [33].

### 2.2. Research Team and Reflexivity

Twelve researchers participated in this study (five women and seven men), all with a doctorate in health sciences and psychology, except two nurses from the clinical field, one with a specialty in mental health (PPH) and another with experience in the out-of-hospital field (JCB). Three researchers (EGB, ATR, PPH) had extensive experience conducting qualitative studies in health sciences. The researchers (ATR, EGB) were responsible for recruiting participants. The remaining authors had no prior contact with any of the

participants. The focus groups were conducted by three researchers specializing in the area of out-of-hospital suicide prevention working in simulation (TPT, PPH, JCB). The positioning of the researchers was established regarding the theoretical framework, their beliefs, previous experience, and personal motivations to participate in the research. The entire team participated in the evaluation of each stage of the research process to reduce researcher bias. The data were triangulated with two external researchers (IVA, AMS).

### 2.3. Participants and Sampling

This study was conducted at a university in the Community of Madrid, Spain. In Spain, the nursing degree consists of 4 academic years. Specifically, the mental health nursing subject in the curriculum is worth 6 ECTS credits and is undertaken in the third year.

Non-probability and purposive sampling were performed by selecting participants based on their ability to provide relevant information in response to the research questions [34].

Inclusion criteria consisted of (a) third-year undergraduate students in nursing and (b) those enrolled in mental health nursing at a Spanish university. Students were selected because of their knowledge and exposure to clinical simulation in mental health. Participation was offered voluntarily to all students. The total possible sample was 48 participants ($n = 48$). Recruitment was carried out until data saturation was obtained [35].

### 2.4. Data Collection

To explore diverse viewpoints, focus groups (FGs) were convened alongside researchers' field notes and participants' written reflective narratives, enriching the analysis with direct observations and personal insights. This qualitative methodology enabled a nuanced and contextualized exploration of the experiences encountered during the simulations. The FGs fostered participant interaction, encouraging the emergence of varied opinions and perceptions. Data collection occurred between October and December 2023, allowing for a comprehensive examination of the subject matter.

Each FG consisted of 10–11 participants, led by a moderator and an observer. The moderator posed questions to which each participant responded by speaking in turn. The observer supported the moderator, identifying key points and taking notes. A topic guide was used, focused enough to collect information about the study area and open sufficient to stimulate participant discussion and interaction (Table 1). However, data collection in qualitative studies is flexible; consequently, during the focus groups, the moderator asked about those areas of interest that participants raised concerning the research question [31,35]. All FGs were audio and video recorded with the prior permission of the participants. The average duration of each FG was 49 min, with four focus groups being conducted, at which time no new information emerged from the data analysis.

**Table 1.** Interview Guide.

| | |
|---|---|
| Post-Clinical simulation phase | 1. Emotions and thoughts of the student handling the situation. |
| | 2. Reasons why the patient is in this situation. |
| | 3. Intervention objectives and strategies implemented. |
| | 4. Risk and protective factors, behavioral precipitants, suicidal history, ambivalence, hopelessness. |
| | 5. Possible alternatives. |

In addition, the students who had participated in the simulation-based experience voluntarily made a written reflective narrative through the Moodle virtual campus, answering the following open questions: *How did you feel during the simulation? What difficulties did you encounter in dealing with the case? What are the points for improvement, and what did you learn? How do you think clinical simulation can help you in the management of a patient with a suicidal crisis?* Thirty-two reflective narratives were collected, with a total of 11,279 written words.

High Fidelity Simulation Procedure

The central objective of the simulation scenario was to train students in the approach, management, and intervention in suicidal crisis in the out-of-hospital context. This training was carried out through experiences based on clinical mental health simulation, where students developed specific nursing competencies to respond effectively to suicidal crises. To this end, two high-fidelity clinical simulation scenarios were developed, taking into account the North American Nursing Diagnosis Association (NANDA), Nursing Outcomes Classification (NOC), and Nursing Interventions Classification (NIC) taxonomy related to each (Table 2) [36]. The NIC taxonomy interventions and the nursing activities served as a guide to discuss the students' performance during the debriefing phase [17,37].

**Table 2.** Scheme simulated scenario, NANDA-I (NANDA-I Taxonomy) Nursing Interventions Classification (NIC) and outcomes (NOC), and related nursing activities for resolution.

| Simulated Scenario 1 | NANDA Diagnose | NIC Intervention | NOC Outcomes | Nursing Activities | |
|---|---|---|---|---|---|
| A 34-year-old male, on our arrival, was on the M-40 bridge. He presented suicidal ideation and the intention to jump. As background, he refers to an argument with his wife. He has two daughters that he has not seen for some time, cocaine consumption, and his van broke down just today when he was going to deliver an order. No one else is on the bridge, only you (nursing students). You are the first to intervene and establish the first contact with the person. | (00150) Suicide risk | (6486) Environmental management: Safety<br><br>(4500) Prevention of substance abuse<br><br>(6340) Suicide prevention | (1408) Self-control of suicidal impulses<br><br>(1904) Risk management: drug use | 1.<br>2.<br>3.<br>4.<br>5.<br>6. | Verbalize if there is suicidal ideation.<br>Seek help when you feel self-destructive feelings.<br>Determine the existence and the degree of suicide risk.<br>Provide information on available resources and programs.<br>Help the patient to identify self-destructive behaviors.<br>Encourage the patient to re-examine negative self-perceptions. |
| Simulated Scenario 2 | NANDA Diagnose | NIC Intervention | NOC Outcomes | Nursing Activities | |
| Middle-aged man threatening to take his own life on the viaduct in Segovia (Madrid). No further information is available at the Emergency Service Center. You arrive as the first unit to intervene. When you come, the person is at high risk of suicide, as he is attached to a railing on the outside of the bridge. It is you (nursing students) who make the first contact. | (00124) Despair<br><br>(00150) Suicide risk | (5230) Increasing coping<br><br>(5270) Emotional support<br><br>(4920) Active Listening<br><br>(6654) Surveillance: security | (1302) Coping with problems<br><br>(1305) Psychosocial modification: life change | 1.<br>2.<br>3.<br>4.<br>5.<br>6.<br>7.<br>8.<br>9. | Show interest in the patient.<br>Show awareness and sensitivity to emotions.<br>Establish the purpose of the interaction, eliminating biases, assumptions, personal concerns, and other distractions.<br>Avoid barriers to active listening.<br>Assist in decision-making.<br>Stay with the patient and provide security.<br>Encourage conversation or crying to decrease the emotional response.<br>Listening to expressions of feelings and beliefs.<br>Help the patient to recognize and express feelings. |

The standardized patients, who portrayed various roles, were experienced educators with expertise in mental health. They meticulously crafted scripts and adeptly enacted

diverse scenarios, offering participants authentic and immersive experiences to refine their clinical skills.

All clinical simulation sessions adhered to the Best Practices outlined by the International Association for Nursing Clinical Learning and Simulation (INACSL), encompassing four key phases [38]: pre-briefing, briefing, development of the simulated scenario, and debriefing. These phases were meticulously executed and overseen by two university professors with expertise in clinical simulation methodology. Notably, during the pre-briefing phase, great emphasis was placed on cultivating a psychologically safe environment following the guidelines proposed by Rudolph et al. [39]. Furthermore, the debriefing phase followed the principles of the good judgment model, facilitating an environment where participants felt empowered to make mistakes, engage in meaningful discussions, and receive constructive criticism. This approach aimed to foster reflective learning experiences wherein participants and instructors could integrate newfound knowledge with existing expertise [39,40].

Through simulated scenarios, participants acquire the ability to identify warning signs, evaluate risk levels, and effectively determine appropriate interventions. Debriefing sessions facilitate reflective thinking and offer constructive feedback, promoting a deeper understanding of the learning experience [41]. Moreover, using a safe learning environment encourages reflection and the application of mental health first-aid techniques [42]. Consequently, participants are empowered to analyze their actions, pinpoint areas for enhancement, and comprehend the potential impact of their interventions in real-world situations.

### 2.5. Data Analysis

The data collection process involved comprehensive verbatim transcriptions of researchers' focus group discussions (FGs), field notes, and students' responses to open-ended questions on the virtual platform. These data were meticulously stored, managed, classified, and organized using the qualitative data analysis software ATLAS.ti 8.0. A thematic analysis was conducted by systematically identifying relevant text segments to address the research inquiry [43]. From these narratives, descriptive elements (codes) were discerned, followed by their categorization based on their inherent meanings and organization into thematic clusters. Ultimately, this iterative process led to identifying overarching themes encapsulating participants' experiences, thereby providing a nuanced depiction of their perceptions.

Three researchers, experts in qualitative research, developed the entire process of obtaining categories and subcategories independently, ending the process with the exchange of both and a consensus on the final decisions of the analysis. In case of divergence of opinions, the theme was identified based on consensus among the research team members.

### 2.6. Ethical Considerations

Ethical authorization to conduct the research was obtained from the Research Ethics Committee of the Instituto de Investigación Sanitaria Fundación Jiménez Díaz (EOH040-257 21_FJD). All participants gave written consent before participating in this study. To ensure anonymity and confidentiality, a code was assigned to each participant in the FGs and 259 reflective narratives (E).

### 2.7. Strictness Criteria

The study followed the Consolidated Criteria for Reporting Qualitative Research (COREQ) [44]. Data triangulation was applied among researchers involved in the analysis, and the analysis process was subject to review by independent researchers to ensure credibility. Transcripts were offered to participants with the opportunity to add any relevant information. Transferability was ensured by a detailed description of the research setting, participants, context, and method. Confirmability was achieved by introducing variability in participants' experiences. Each researcher conducted the reading and analysis independently, contrasting and then agreeing on emerging themes and subthemes [45].

### 3. Results

*3.1. Demographic Data*

Of the total number of nursing students who met the criteria, 45 (93.8%) participated (Table 3). Only 3 possible participants out of 48 did not attend the simulation for lack of time, work shifts, or other reasons. Most participants were female 41 (91.1%), compared to 8.9% who were male (8.9%). They ranged in age from 20–27 years (mean age = 21.3; SD = 2.29%). The discrepancy between students and genders is mainly because most students enrolled in the profession in Spanish universities are women. The participants belonged to the third year of nursing.

**Table 3.** Demographic Information for Students as Mean (SD) or Count (%).

| Variables | *N* (%) |
|---|---|
| Age (Mean, SD) | 21.36 (2.29) |
| **Gender** | |
| Female | 41 (91.1%) |
| Male | 4 (8.9%) |

*3.2. Themes*

From the lived experiences, three thematic blocks with their categories were identified (Table 4): (a) management and handling of emotions, (b) identification of the reasons for suicide, and (c) intervention.

**Table 4.** Themes and categories.

| | Themes (T) | Categories |
|---|---|---|
| T 1 | Management and handling of emotions | Fear, uncertainty, empathy, trust, learning, specialty. |
| T 2 | Indicators of suicidal behavior | Risk factors and protective factors, precipitating factors of suicidal behavior, suicidal history, and ambivalence. |
| T 3 | Suicidal crisis intervention | Establishment of the relationship, intervention strategies and alternatives. |

Theme 1: Management and handling of emotions.

Participation generated palpable expectation and nervousness, indicating a significant emotional immersion: "*I was expectant and nervous.*" E1

The participants expressed as a lived experience, the presence of uncertainty as a challenging element since they recognized the lack of control over the situation:

"*The uncertainty, isn't it? Because in the end, the situation doesn't depend on you. . .*" FG3

Some participants expressed a strong sense of empathy for the simulated patient:

"*I don't know. . . as if he was always telling bad things, you empathized a lot with him. . . and you said fuck. . . and you wanted to find something good for him to get out of that. . . out of that loop from which he wants to commit suicide. . . I don't know. . .*" FG2: "*I have felt the desperation they suffer, the inability to cope with situations.*" FG1

The presence of fear was highlighted, especially concerning sensitive topics such as suicide: "*I felt fear because suicide seems to me a very important topic.*" E2 But just the simulation was revealed as an effective tool to overcome such fear, evidenced by statements indicating greater confidence and preparedness:

"*My fear of these situations has been removed, and I find myself much more prepared*" E15 "*I gained confidence by remembering the points explained earlier.*" E4

Even for some students, it motivated them to consider specialization in mental health, demonstrating a positive impact on career aspirations. In some cases, there was a perceptual shift, with participants initially disinterested in mental health, expressing renewed interest and enthusiasm.

> *"I have been very motivated thinking about possibly doing the mental health specialty."* E27 *"I thought I didn't like mental health, but since the simulation, it has caught my attention, and I am looking forward to it."* E31

The simulation was perceived as a challenge that facilitated deep reflection, creating a meaningful bridge between academic theory and practical application:

> *"It has been challenging, helping us to reflect and having a link between theory and practice."* E20

> *"I have learned the keys to interventions for these patients."* E22

Including real actors as patients were identified as a critical factor in increasing the realism of the simulation in this type of patient. The student 19 added:

> *"It has been fundamental to have a real actor as a patient. It has made the simulation much more realistic."* E19

Theme 2: Indicators of suicidal behavior

The reflective narratives submitted by participating students underscore the significance placed on identifying motives and factors associated with patients exhibiting suicidal intentions.

> *"It is important to identify why the person is in that situation."* E1. This statement is supported and enriched by the comments collected during the FGs. In one of the groups, various aspects that could contribute to the patient's situation were explored:

> *"The economic issue, the family issue, the divorce issue, the relationship with the daughters, all these are risk factors."* FG1

This observation underscores the diversity of factors that students consider crucial to understanding the complexity of situations related to suicidal intent. Additionally, protective factors also emerged as critical elements in the participants' reflections:

> *"...it is an indicator, it is a protective factor of beliefs."* FG4

This recognition suggests that students are not only focused on the risk aspects, but also value the identification of elements that can mitigate the risk of suicide in patients. Suicidal career experiences, according to participants' narratives, are revealed as a complex cumulative process. One participant expressed:

> *"The suicidal career can be very varied, but such events accumulate that I accumulate, and there comes a time when I can't take it anymore, and that drop is predisposed."* FG3

This story illustrates the diversity of events that, over time, can contribute to the development of suicidal ideation, emphasizing the importance of considering the cumulative process in assessment and intervention. Concerning family relationships, they provided additional insights into the ambivalence experienced by some participants. The complexity of family relationships was described, especially in the context of the maternal Figure 1.

> *"The wife... the daughters... have been a bit ambivalent, right? In the beginning, it seemed like... then it seemed like... with you, it seemed like."* FG1

The creation of ambivalence in decision-making related to a suicidal career was a recurring theme:

> *"Creating ambivalence, yes it can be all and I at the end for them to decide and make that decision to take another path."* FG2

*"Once the life hitch was identified, it has been easier to establish ambivalence."* E9

This finding underscores the intricate interplay of psychological and emotional factors that can contribute to the decision to take suicidal action, emphasizing the necessity for specialized interventions aimed at resolving internal conflicts.

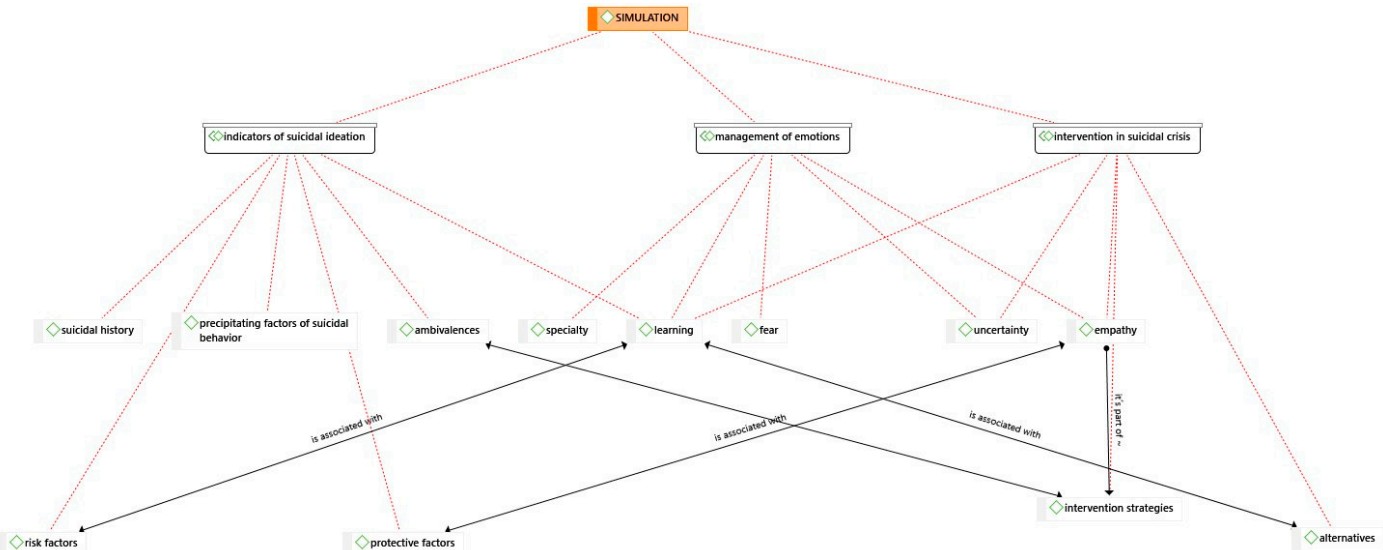

**Figure 1.** Qualitative Data Analysis.

Theme 3: Suicide crisis intervention

As previously evidenced, students, through the simulation, have gained a deeper understanding of the necessary approach, as reflected in the statement:

*"We have learned how the approach to the simulation should be, following the steps."* E12

They recognize the importance of establishing a solid initial connection, as expressed by a FG participant:

*"The connection, what's the first thing to do? Hook up with the person; otherwise, you won't have anything."* FG4

This statement suggests an awareness of the proper sequence of steps to establish effective relationships. Creating an appropriate rapport is essential to ensure the safety of the self and simulated patient.

*"I introduced myself, tried to get him to talk to me, and got down to his level."* FG2

Students believe that they have followed protocols to safeguard safety:

*"I position myself as a safety measure to ensure that the person does not catch me at a given moment and throw me away, but I put myself at their level."* E12

The intervention reveals the implementation of specific strategies

*"My partners and I did a pretty thorough approach. . .using relaxation techniques. . .looking for alternatives."* E4

This approach suggests the practical application of therapeutic techniques and an active search for solutions. However, uncertainty and concern about the effectiveness of the strategies employed arise, as evidenced by the statement:

*"Sometimes I've felt like it was like what the two partners said a little bit, you're left kind of thinking I don't know if what I'm going to say now will be impactful enough for me to pay attention."* FG2

The importance of offering viable alternatives is highlighted in both the individual narratives and the FGs. One participant noted:

*"I think the main thing is to accompany them, listen to them and make them feel supported. . .give alternatives that we can give him because he is not seeing them."* E5

The need to provide concrete options and emotional support to the patient is emphasized. In addition, communication of the availability of other alternatives is evident.

*"I told him that there were other alternatives to quitting that, even though he tried it once, it doesn't mean it's the only one, that he can try again."* FG3

These results underscore the intricate nature of the intervention, emphasizing the importance of employing practical strategies and considering various alternatives when addressing simulated mental health scenarios. Figure 1 delineates several key themes: 1. Management and Handling of Emotions, encompassing five subthemes (Empathy, Uncertainty, Specialty, Fear, Learning), 2. Indicators of Suicidal Behavior, consisting of five subthemes (Learning, Ambivalences, Precipitating Factors of Suicidal Behavior, Suicidal History, and Risk Factors), and 3. Suicidal Crisis Interventions with five subthemes (Alternatives, Intervention Strategies, Uncertainty, Empathy, and Learning).

These themes are interconnected and influenced by the use of the ATLAS.ti program for simulation analysis, which, as indicated by the findings, may have an impact on the presented suicidal risk.

## 4. Discussion

This study aimed to explore the perceptions of nursing students concerning the approach, management, and intervention in suicidal crises through experiences based on mental health clinical simulation. The use of clinical simulation in the field of mental health nursing, especially in suicide crisis intervention and management, stands out as an innovative and essential pedagogical approach in student education, supported by previous studies [46,47]. The results of the study are aligned with prior research, evidencing that simulation experiences provide students with the unique opportunity to confront, manage, and make decisions in a controlled and safe environment, contributing significantly to improving their confidence and competence in suicidal crisis intervention [30]. The development of fundamental skills, such as meaningful connection with patients through the practice of active listening, empathy, and effective use of verbal and nonverbal language, was highlighted as a crucial aspect of establishing therapeutic relationships and providing support to people in suicidal crisis [18,48]. Likewise, the identification and assessment of risks and warning signs during the simulation allowed students to assess risk factors associated with suicide effectively, consolidating their preparedness for real situations [24].

The significance of understanding the underlying motives for suicide was emphasized by participants, highlighting their recognition of the importance of delving into the reasons behind suicidal intentions. Moreover, participants acknowledged the criticality of identifying protective factors, demonstrating a balanced perspective beyond solely focusing on risk factors. Furthermore, the exploration of suicide as a complex, cumulative process underscored the necessity of considering developmental trajectories over time in assessment and intervention. These findings reflect a mindful, multidimensional approach toward identifying motives and factors in individuals with suicidal intent, alongside a profound appreciation for the importance of establishing a solid initial connection. Moreover, the results underscore the crucial role of thorough assessment and the consideration of diverse elements in clinical practice and health professional training. They emphasize the significance of simulations in mental health professionals' training [49], not only for honing practical skills, but also for influencing participants' personal perceptions and career aspirations. Incorporating emotional elements and contextual realism emerges as pivotal in maximizing the positive impact of simulations on professional development. This aligns with existing research suggesting that simulated patient training involving actors enables the attainment of realism across various pathological scenarios, thereby enhancing educational interventions in novel situations [50,51].

Through simulated scenarios, participants gained invaluable insights into suicide crisis intervention, deepening their understanding of the requisite approach. Recognizing the pivotal role of establishing a solid initial connection, participants demonstrated an acute awareness of the sequential steps necessary to foster effective relationships. The practical application of therapeutic strategies, such as relaxation techniques and active problem-solving, exemplified translating theoretical knowledge into action. Nonetheless, participants expressed some uncertainty regarding the efficacy of these strategies, underscoring the ongoing need for skill development [16]. The management and regulation of emotions during simulations provided a profoundly immersive experience characterized by anticipation, nervousness, and even fear, highlighting the intensity of the simulated situations. Despite grappling with feelings of uncertainty and a perceived lack of control, participants emerged from the simulations with heightened confidence and a sense of preparedness, indicative of the effectiveness of simulation in alleviating fear [10]. While participants acknowledged the authenticity of the simulated scenarios, confronting challenges within a controlled environment fortified their ability to navigate crises safely and effectively in their professional endeavors [52].

In the out-of-hospital suicide risk simulation, nursing students play a crucial role in identifying and managing suicidal ideation. This involves recognizing verbal and non-verbal signs of suicidal ideation, being alert to self-destructive behaviors, and thoroughly assessing suicide risk. Additionally, students must provide information about available resources and offer emotional support, fostering hope and self-efficacy in the patient. These skills are practiced through practical roles and case discussions in simulations, preparing students to face suicide risk situations in real-world settings and provide safe and effective care.

Therefore, the results of this study highlight the satisfaction of nursing students with the simulation experience in suicidal crisis management. Beyond their improved competencies and confidence, the simulation has had a transformative impact on the participants' career aspirations, motivating them to consider specialization in mental health. This perceptual shift involves not only a deeper appreciation of the complexity of suicide crisis intervention, but also overcoming previous stigmas associated with mental health [53]. Thus, simulation emerges not only as an effective tool for practical training, but also as a positive catalyst in the perception and approach of future nursing professionals to the vital area of mental health.

## 5. Strengths and Limitations of the Study

The strengths of this study are twofold. Firstly, it offers an innovative perspective on the necessity to revamp the traditional university teaching model, advocating for developing new lifelike scenarios to enhance knowledge acquisition. Secondly, the study highlights the value of providing nursing students with comprehensive training before entering the workforce. This training not only equips them with a robust knowledge base for their future practice, but also bolsters their confidence in effectively managing patients at risk of suicide.

Regarding the study's limitations, it was carried out with nursing students from a Spanish university. Therefore, the results cannot be generalized to other nursing students from other universities. The present study reflects the perceptions of nursing students in different high-fidelity clinical simulation scenarios based on a patient at risk of suicide. It is interesting to extend the qualitative research on a larger scale with new scenarios of clinical mental health cases in simulation and associated studies with a quantitative methodology to evaluate the degree of effectiveness of the educational intervention.

## 6. Conclusions

The results of this study indicate that nursing students face challenges in approaching clinical mental health simulation due to a lack of prior exposure. Simulation in suicide intervention is presented as an effective tool to overcome these difficulties by providing a controlled environment. The results show that, despite initial insecurity, students experi-

ence a positive change in their attitudes, expressing more significant interest in the mental health specialty and reducing the associated stigma. In conclusion, clinical simulation improves skills and significantly changes perceptions of mental health.

**Author Contributions:** Conceptualization, P.D.P.-H., A.T.-R., A.M.-S., E.C.-S., R.J.-V. and E.G.-C.B.; Data curation, T.P.-T., S.G.-F. and I.S.-A.; Formal analysis, P.D.P.-H., A.T.-R., T.P.-T. and J.C.-Y.; Funding acquisition, E.C.-S.; Investigation, P.D.P.-H., E.C.-S. and T.P.-T.; Methodology, P.D.P.-H., J.C.-Y. and R.J.-V.; Project administration, P.D.P.-H., E.C.-S. and R.J.-V.; Resources, A.T.-R., A.M.-S. and J.L.S.-G.; Software, A.M.-S., E.C.-S., T.P.-T. and J.L.S.-G.; Supervision, A.M.-S., T.P.-T., T.S.-S., R.J.-V. and E.G.-C.B.; Validation, A.M.-S., T.P.-T. and J.C.-Y.; Visualization, J.L.S.-G. and R.J.-V.; Writing—original draft, P.D.P.-H. and A.M.-S.; Writing—review and editing, A.T.-R., E.C.-S., J.L.S.-G., S.G.-F., I.S.-A., T.S.-S., R.J.-V. and E.G.-C.B. All authors have read and agreed to the published version of the manuscript.

**Funding:** This research received no external funding.

**Institutional Review Board Statement:** The study was conducted following the Declaration of Helsinki and approved by the Institutional Review Board (or Ethics Committee) of Instituto de Investigación Sanitaria Fundación Jiménez Díaz (protocol code EOH040-21_FJD).

**Informed Consent Statement:** Informed consent was obtained from all student nurses involved in the study.

**Data Availability Statement:** Data are contained within the article.

**Public Involvement Statement:** Participants in the study were the learner nurses from the University. Consent was obtained from all participants to use their personal stories as data.

**Guidelines and Standards Statement:** This manuscript was drafted against the Consolidated criteria for reporting qualitative research (COREQ) for qualitative research.

**Acknowledgments:** The authors acknowledge all the students participating in this study.

**Conflicts of Interest:** The authors declare no conflicts of interest.

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
