# Peer review of "Mental Health Nursing Student’s Perception of Clinical Simulation about Patients at Risk of Suicide: A Qualitative Study"

_nursrep, doi:10.3390/nursrep14010049_

Round 1
Reviewer 1 Report
Comments and Suggestions for Authors
Dear authors ;
Upon thorough review, we believe your article has great potential but would benefit from some refinements to reach its fullest impact. Specifically, we encourage you to delve deeper into the phenomenological framework that underpins your study. It would be enriching to clarify which phenomenological perspective you are aligning with, such as Heidegger's interpretive phenomenology or Husserl's descriptive phenomenology. These theoretical foundations could greatly enhance the depth and clarity of your findings.
It is suggested to use the Consolidated Criteria for Reporting Qualitative Research (COREQ)
Additionally, it would be beneficial to specify the context of suicide prevention you are addressing - whether it is within an inpatient (intrahospital) or outpatient (extrahospital) setting. This clarification will provide readers with a more precise understanding of the study's applicability and relevance.
Moreover, we recommend that you elaborate on the specific nursing actions and interventions identified in the hypothetical cases within your simulations. For instance, it would be insightful to illustrate how nursing students verbalize the recognition of suicidal ideation, seek help for self-destructive feelings, assess the degree of suicide risk, provide information on resources, identify self-destructive behaviors, and encourage patients to re-examine negative self-perceptions. Detailing how these actions are evidenced in simulations could offer valuable practical insights for both educators and practitioners in mental health nursing.
To enhance your manuscript further, consider incorporating concepts like 'lived experience' and 'being-in-the-world' which are commonly utilized in phenomenological research within nursing, to provide a richer narrative of the students' experiences.
Your dedication to advancing knowledge in this critical area is truly commendable, and we are excited about the possibility of publishing your enhanced work. Should you have any questions or require further clarification on the suggested revisions, please do not hesitate to contact us.
Reviewer 2 Report
Comments and Suggestions for Authors
This an important and contemporary topic on how clinical simulation of suicidality is viewed by nursing students. I note that the authors have stated that the study was conducted in accordance with the Consolidated Criteria for Reporting 234 Qualitative Research (COREQ). Below are requests to address outstanding reporting criteria.
Can the author please confirm if the moderator and observers for each of the focus groups were also authors and if so please state which authors.
Please state what the researcher’s credentials are of who conducted the focus groups and performed the qualitative analysis (e.g. PhD MD).
Please state what the researcher’s occupation was at the time of the study and their gender.
Please state what experience or training the researcher has.
Please state if a relationship was established prior to commencing the study.
Please state participant knowledge of the interviewer. What did the participants know about the researcher? e.g. personal goals, reasons for doing the research
Please state interviewer characteristics. What characteristics were reported about the interviewer/facilitator? e.g. Bias, assumptions, reasons and interests in the research topic
Please state if transcripts were returned to participants for comment and/or correction?
Please state if participants provided feedback on the findings?
Lines 88-89: Please amend use of citations as they appear to be doubled up for this sentence.
Lines 155, 159, 212, 231 plus multiple instances throughout the qualitative analysis: please amend if ‘GF’ was intended to be ‘FG’
Line 184: Please write in full the first time they are used what the following mean NANDA, NOC NIC
Reviewer 3 Report
Comments and Suggestions for Authors
Dear Authors:
Thank you for allowing me to review this manuscript. I think the manuscript is interesting and requires only minor changes and some clarification. I am enclosing some comments with the aim of improving this manuscript.
- Participants and sampling: Since all students were offered participation, if any, it should be stated how many students declined the offer to participate and the reasons if possible. Also who did the recruitment and the relationships between the participants and the researchers.
In qualitative research the context is important, you could provide information on which university the participants were studying at, as well as some information on the mental health subject at that university, as there may be variations in curricula between universities and countries.
Standardise the acronym for focus groups (FG or GF).
Since you are adhering to the Consolidated Criteria for Reporting Qualitative Research (COREQ) standards, Domain 1: Research team and reflexivity should appear more clearly (Which author/s conducted the interview or focus group? What were the researcher's credentials? E.g. PhD, MD What was their occupation at the time of the study? Was the researcher male or female? What experience or training did the researcher have?. Also, as mentioned above, the Relationship with participants should be explained.
-In the triangulation process, who were the independent researchers? Were they the same members of the research team or external researchers?
-Were the participants offered the transcripts? Were the results of the analysis checked with some of the participants?
-The results are somewhat confusedly presented. It would have been interesting to present the composition by focus groups (number of participants in each group, general characteristics), in order to be able to assess whether there were differences in the groups, which is desirable as it adds richness to the discourse. In this way the verbatim could be identified by participant and focus group. Although it is impossible to provide a table with all participants, age and gender, for example, could be added to the verbatim identification. Finally, Figure 1 has a format that needs to be improved in order to be able to read the labels and the relationships between them.
Good work.Best regards.
